# Efficient Off-Policy Learning for High-Dimensional Action Spaces

**Fabian Otto**[*]
Microsoft Research
fabian.otto@microsoft.com

**Philipp Becker**
Karlsruhe Institute of Technology

**Ngo Anh Vien**
Bosch Center for Artificial Intelligence

**Gerhard Neumann**
Karlsruhe Institute of Technology

## Abstract

Existing off-policy reinforcement learning algorithms often rely on an explicit state-action-value function representation, which can be problematic in high-dimensional action spaces due to the curse of dimensionality. This reliance results in data inefficiency as maintaining a state-action-value function in such spaces is challenging. We present an efficient approach that utilizes only a state-value function as the critic for off-policy deep reinforcement learning. This approach, which we refer to as Vlearn, effectively circumvents the limitations of existing methods by eliminating the necessity for an explicit state-action-value function. To this end, we leverage a weighted importance sampling loss for learning deep value functions from off-policy data. While this is common for linear methods, it has not been combined with deep value function networks. This transfer to deep methods is not straightforward and requires novel design choices such as robust policy updates, twin value function networks to avoid an optimization bias, and importance weight clipping. We also present a novel analysis of the variance of our estimate compared to commonly used importance sampling estimators such as V-trace. Our approach improves sample complexity as well as final performance and ensures consistent and robust performance across various benchmark tasks. Eliminating the state-action-value function in Vlearn facilitates a streamlined learning process, yielding high-return agents.

## 1 Introduction

*Reinforcement learning* (RL) has emerged as a powerful paradigm for training intelligent agents through interaction with their environment (Mnih et al., 2013; Silver et al., 2016). On-policy and off-policy are the two primary approaches within model-free RL On-policy methods rely on newly generated (quasi-)online samples in each iteration (Schulman et al., 2017; Otto et al., 2021), whereas off-policy methods leverage a replay buffer populated by a behavior policy (Abdolmaleki et al., 2018; Haarnoja et al., 2018; Fujimoto et al., 2018). Although on-policy methods can compensate for off-policy data to some extent via importance sampling (Espeholt et al., 2018), they cannot exploit it fully. To harness the full potential of off-policy data, off-policy methods traditionally focus on learning state-action-value functions (Q-functions) as critics (Degris et al., 2012).

In this work, we present *Vlearn*, a off-policy policy gradient method that exclusively utilizes V-functions. With Vlearn, we do not seek to replace existing methods, but rather to offer a new perspective and practical insights for deep off-policy RL by demonstrating the general feasibility and benefits of such a setting. Unlike commonly used Q-function based methods, this approach does not require learning over the joint state-action space, making it less susceptible to the curse of dimensionality and more effective at updating the critic regardless of the actions taken when acting in the environment. Consequently, Vlearn naturally demonstrates better suitability for high-dimensional action spaces. While existing V-function methods, such as V-trace (Espeholt et al.,

---

[*]Work partially done at Bosch Center for Artificial Intelligence and University of Tübingen

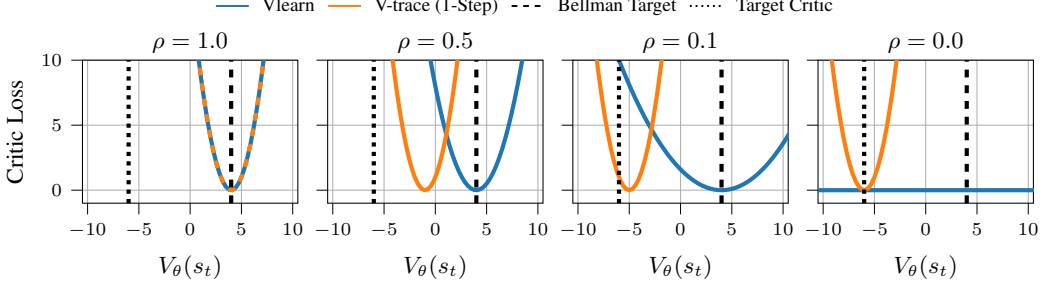

Figure 1: To provide intuition on the differences between Vlearn and V-trace, we consider the following example for different importance ratios $\rho$. Suppose for a state $s_t$ the Bellman target is $r(s, a) + \gamma V_{\bar{\theta}}(s_{t+1}) = 4$, the target critic predicts $V_{\bar{\theta}}(s_t) = -6$ and we plot the loss for values of $V_\theta(s)$ for Vlearn and V-trace. For on-policy samples ($\rho = 1.0$), both losses are the same. However, for increasingly off-policy samples ($\rho \to 0$), we see how V-trace increasingly relies on the target critic, shifting the optimal value towards it. Vlearn, on the other hand, simply reduces the scale of the loss and thus the importance of the sample, making Vlearn more robust to errors in the target critic.

2018), attempt to increase the usability of stale data in on-policy methods, we find that they are less effective in a completely off-policy context. These methods typically focus on reweighting Bellman targets, but prior research (Mahmood et al., 2014) in the linear domain indicates that applying importance sampling to the entire Bellman error is more advantageous. Inspired by these findings, we demonstrate that this objective can be recovered in a deep RL setting as an upper bound on the original naive Bellman error. This upper bound can be derived using Jensen's inequality and retains the same minimum. This loss shifts the importance weights from the Bellman targets to the objective, simplifying V-function updates and enhancing the stability of learning a V-function from off-policy data, which posed significant challenges in earlier methods. Our analysis in the bandit setting shows that this Bellman error estimator has lower variance compared to the estimators from common deep RL importance sampling schemes like V-trace (Espeholt et al., 2018).

Crucially, we propose a practical algorithm that leverages this objective and improves the stability of policy learning by using the trust region update from Otto et al. (2021) and other commonly used techniques from off-policy deep RL, such as twin critics and delayed policy updates (Fujimoto et al., 2018), `tanh` squashing (Haarnoja et al., 2018), and advantage normalization (Schulman et al., 2017).

Our experiments demonstrate the relevance and importance of these design choices in achieving high performance. Notably, by removing the dependence on the Q-function our method is particularly well suited for environments with complex action spaces, such as the challenging MyoSuite (Vittorio et al., 2022) and dog locomotion tasks in *DeepMind control* (DMC) (Tassa et al., 2018), which most standard off-policy actor-critic methods cannot solve.

## 2 RELATED WORK

To enhance sample efficiency in RL, off-policy algorithms aim to leverage historical data stored in a replay buffer (Lin, 1992). While deep Q-Learning-based methods (Mnih et al., 2013; Van Hasselt et al., 2016; Wang et al., 2016; Hessel et al., 2018) excel in efficiently learning from discrete action spaces, they are often not directly applicable to continuous problems. For such problems, off-policy actor-critic methods optimize a policy network that generates continuous actions based on the state-action-value function estimator (Degris et al., 2012; Zhang et al., 2019). This approach accommodates both deterministic (Lillicrap et al., 2015; Fujimoto et al., 2018) and probabilistic policies (Haarnoja et al., 2018; Abdolmaleki et al., 2018).

In the on-policy setting, trust-region methods (Schulman et al., 2017; 2015; Otto et al., 2021) have proven effective in stabilizing the policy gradient. However, these methods have not received the same level of attention in the off-policy setting. Past efforts have focused on training the value function with off-policy data (Gu et al., 2016) or extended trust region policy optimization (Schulman et al., 2015) to the off-policy setting (Nachum et al., 2018; Meng et al., 2022). While Wang et al. (2017), for instance, employs a more standard approach by combining off-policy trust regions with additional

advancements, such as Retrace (Munos et al., 2016) and truncated importance sampling, it is still not able to compete with modern actor-critic approaches. Peng et al. (2019), on the other hand, provide a different perspective and split policy and value-function learning into separate supervised learning steps. Generally, standard off-policy trust region methods often fall behind modern actor-critic approaches (Haarnoja et al., 2018; Fujimoto et al., 2018) and cannot achieve competitive performance. However, *maximum aposteriori policy optimisation* (MPO) (Abdolmaleki et al., 2018) can be seen as a non-classical trust region method. Its EM-based formulation offers more flexibility, with trust regions akin to optimizing a parametric E-step without an explicit M-step.

While on-policy trust region methods estimate the less complex state-value function, most off-policy methods rely on estimating state-action-value functions to learn from replay buffer data (Haarnoja et al., 2018; Fujimoto et al., 2018; Abdolmaleki et al., 2018). Methods exclusively utilizing state-value functions must employ importance sampling to address the distributional discrepancy between target and behavior policies. However, this has been primarily studied in the context of asynchronous on-policy settings, where the policy distributions differ only slightly (Espeholt et al., 2018; Luo et al., 2020). In particular, V-trace (Espeholt et al., 2018) aims to compensate for a limited distributional shift due to asynchronous computations and is not designed for a full off-policy setting, such as in this work. Further, such an off-policy correction is computationally expensive as it relies on storing and processing trajectories. Moreover, the truncation technique used in importance weight calculations introduces a bias (Espeholt et al., 2018). Consequently, the implementation of trajectory-based Bellman target estimators would exacerbate the already pronounced issue of bias propagation.

In the linear case, the effect of varying the placement of the importance weight in the off-policy case has already been studied. Applying importance weighting to the entire Bellman error, rather than just the Bellman target, yields better performance and may even lead to reduced variance. This approach is naturally used in methods such as off-policy TD(0) (Sutton & Barto, 2018), and further supported by Mahmood et al. (2014); Dann et al. (2014); Graves & Ghiassian (2022) on its effectiveness in the linear setting. However, translating these findings to deep RL presents a challenge, as insights from linear models are often not directly applicable (Mnih et al., 2013; Lillicrap et al., 2015).

## 3 DEEP OFF-POLICY LEARNING FROM STATE-VALUE FUNCTIONS

We seek to determine an optimal policy $\pi(a|s)$ within the framework of a *Markov decision process* (MDP), which is defined by a tuple $(\mathcal{S}, \mathcal{A}, \mathcal{T}, r, \rho_0, \gamma)$. Here, both the state space $\mathcal{S}$ and action space $\mathcal{A}$ are continuous. The transition density function $\mathcal{T} : \mathcal{S} \times \mathcal{A} \times \mathcal{S} \to \mathbb{R}^+$ is defined as a function mapping from the current state $s_t \in \mathcal{S}$ and action $a_t \in \mathcal{A}$ to the probability density of transitioning to the subsequent state $s_{t+1} \in \mathcal{S}$. The initial state distribution's density is denoted as $\rho_0 : \mathcal{S} \to \mathbb{R}^+$. The reward attained from interactions with the environment is determined by the function $r : \mathcal{S} \times \mathcal{A} \to \mathbb{R}$, and the parameter $\gamma \in [0, 1)$ represents the discount factor applied to future rewards. Our primary objective is to maximize the expected cumulative discounted reward $G_t = \mathbb{E}_{\mathcal{T}, \rho_0, \pi} \left[ \sum_{k=t}^{\infty} \gamma^{k-t} r(s_k, a_k) \right]$. Most popular off-policy actor-critic methods (Haarnoja et al., 2018; Fujimoto et al., 2018; Abdolmaleki et al., 2018) now aim to find a policy that maximizes the cumulative discounted reward by making use of a learnable state-action value estimate $Q_\theta^\pi(s, a) = \mathbb{E}_\pi [G_t | s_t = s, a_t = a]$. To train this estimator, they rely on a dataset $\mathcal{D} = \{(s_t, a_t, r_t, s_{t+1})_{t=1...N}\}$ and a behavioral policy $\pi_b(\cdot|s)$ responsible for generating this dataset. Typically, $\mathcal{D}$ takes the form of a replay buffer (Lin, 1992), with the corresponding behavior policy $\pi_b$ being a mixture of the historical policies used to populate the buffer. Training the state-action value function then usually involves temporal difference learning (Sutton, 1988; Watkins & Dayan, 1992), with updates grounded in the Bellman equation (Bellman, 1957). The objective commonly optimized is

$$\mathbb{E}_{(s_t, a_t) \sim \mathcal{D}} \left[ \left( Q_\theta^\pi(s_t, a_t) - \left( r(s_t, a_t) + \gamma \mathbb{E}_{s_{t+1} \sim \mathcal{T}(\cdot|s_t, a_t), a_{t+1} \sim \pi(\cdot|s_{t+1})} Q_{\bar{\theta}}(s_{t+1}, a_{t+1}) \right) \right)^2 \right], \quad (1)$$

where $Q_{\bar{\theta}}(s, a)$ is a frozen target network. Furthermore, to mitigate overestimation bias, most methods employ two state-action value functions (Fujimoto et al., 2018). To maintain the stability of this approach, the target network's weights are updated via polyak averaging $\bar{\theta} \leftarrow \tau\theta + (1 - \tau)\bar{\theta}$.

### 3.1 IMPORTANCE SAMPLING ESTIMATORS FOR THE BELLMAN ERROR

Instead of using the state-action function $Q_\theta^\pi(s_t, a_t)$, an alternative approach is to work solely with the state-value function $V_\theta^\pi(s_t)$. This base estimator can be trained by minimizing the following loss

function using importance sampling

$$L_{\text{base}}(\theta) = \mathbb{E}_{s_t \sim \mathcal{D}} \left[ \left( V_\theta^\pi(s_t) - \mathbb{E}_{a_t \sim \mathcal{D}, s_{t+1} \sim \mathcal{T}(\cdot|s_t, a_t)} \left[ \frac{\pi(a_t|s_t)}{\pi_b(a_t|s_t)} \bar{V} \right] \right)^2 \right]. \qquad (2)$$

where $\bar{V} = r(s_t, a_t) + \gamma V_{\bar{\theta}}(s_{t+1})$ with target network weights $\bar{\theta}$. Directly optimizing Equation 1 or Equation 2 can be difficult due to the double sampling problem (Zhu & Ying, 2020). Approximating the inner expectation with one Monte Carlo sample yields a high variance. Getting a more reliable estimate requires multiple samples, which implies either multiple action executions per state or occurrences of the same state in the replay buffer, an unrealistic assumption for most RL problems. A key distinction from the Q-function based objective (Equation 1) is the introduction of the importance weight, which compensates for the discrepancy between the behavior distribution $\pi_b(\cdot|s)$ and the current policy $\pi(\cdot|s)$. Unlike the Q-function that updates estimates solely for the selected action, the V-function does not depend on specific actions and implicitly updates estimates for all actions. The difference between the current and the data-generating policy must therefore be taken into account.

Equation 2 is closely related to the 1-step V-trace estimate (Espeholt et al., 2018). It can be seen as a naive version of V-trace because a large difference between the target and behavior policies may result in importance weights approaching either zero or infinity, consequently impacting the Bellman optimization target for the state-value function $V_\theta^\pi(s_t)$. While truncated importance weights can prevent excessively large values for the Bellman target, the importance ratios and Bellman target may still approach values close to zero. V-trace circumvents this issue by effectively interpolating between the Bellman target and the value function estimate in the 1-step case

$$L_{\text{V-trace}}(\theta) = \mathbb{E}_{s_t \sim \mathcal{D}} \left[ \left( V_\theta^\pi(s_t) - \left( \mathbb{E}_{a_t \sim \mathcal{D}, s_{t+1} \sim \mathcal{T}(\cdot|s_t, a_t)} \left[ (1 - \rho_t) V_{\bar{\theta}}(s_t) + \rho_t \bar{V} \right] \right) \right)^2 \right], \qquad (3)$$

where $\rho_t = \min(\pi(a_t|s_t)/\pi_b(a_t|s_t), \epsilon_\rho)$ are the truncated importance weights with a user-specified upper bound (typically $\epsilon_\rho = 1$). This objective can then also be extended for off-policy corrections of n-step returns (Espeholt et al., 2018). Yet, this formulation has two main drawbacks. Since executing multiple actions for the same state is impractical, or even impossible, V-trace approximates the inner expectation with just one action sample, resulting in potentially undesirable high variance estimates. Moreover, while the V-trace reformulation avoids optimizing the value estimate toward zero for small importance ratios, the interpolation now optimizes it increasingly toward the current (target) value function (see Figure 1). This interpolation also leads to a shift of the optimum, which means for samples with small importance ratios the value function is barely making any learning progress and maintains its current estimate. It is important to note that only the optimum changes; the scale of the loss function, and thus the scale of the gradient, remains the same for all importance ratios. In the original work, this issue is less pronounced, as V-trace provides off-policy corrections for asynchronous distributed workers of an on-policy method. In this setup, the policy distributions are expected to stay relatively close, mitigating the impact of any potential divergence. However, in a complete off-policy setting, such as in our case, the samples in the replay buffer and the current policy deviate significantly faster and stronger from each other. The size of the replay buffer, learning speed, or entropy of different policies can all influence the importance ratio.

## 3.2 OFF-POLICY STATE-VALUE FUNCTION LEARNING

To address the above-mentioned issues of V-trace, we propose a more effective approach to training the value function by following Mahmood et al. (2014); Dann et al. (2014) and shifting the importance ratio in front of the loss function by extending their results to the general function approximation case. We show that both loss functions, i.e., Equation 2 and Equation 4 share the same optimum.

**Theorem 1** *Consider the following loss, minimized with respect to the V-function's parameters $\theta$*

$$L_{WIS}(\theta) = \mathbb{E}_{(s_t, a_t) \sim \mathcal{D}, s_{t+1} \sim \mathcal{T}(\cdot|s_t, a_t)} \left[ \frac{\pi(a_t|s_t)}{\pi_b(a_t|s_t)} \left( V_\theta^\pi(s_t) - \bar{V} \right)^2 \right]. \qquad (4)$$

*This objective serves as an upper bound to the importance-weighted Bellman loss (Equation 2). Furthermore, this upper bound is consistent; that is, a value function minimizing Equation 4 also minimizes Equation 2.*

The first statement's proof relies on Jensen's Inequality, while the proof of the second statement is an extension of Neumann & Peters (2008). The complete proofs are provided in Appendix A.1 and Appendix A.2, respectively.

This bound closely resembles the standard state-action-value loss functions used in other off-policy methods but is based on the action-independent state-value function. It just introduces importance weights in front of the loss to account for the mismatch between behavior and policy distribution. Importantly, evaluating Equation 4 becomes straightforward using the provided replay buffer $\mathcal{D}$.

**Connection to Linear Off-Policy Methods.**   Interestingly, optimizing this upper bound recovers the linear off-policy TD(0) update (Sutton & Barto, 2018, Chapter 11.1) for the non-linear setting. While the intuitive objectives for linear off-policy TD(0) might appear to be either Equation 2 or Equation 3, the actual update results from optimizing Equation 4. Mahmood et al. (2014); Dann et al. (2014) further demonstrated that Equation 2 and Equation 4 can be connected to ordinary and *weighted importance sampling* (WIS) in the linear setting, respectively. These findings can be applied for linear value function estimation methods such as LSTD (Bradtke & Barto, 1996) by employing the linear version of the loss from Equation 4, thereby obtaining similar theoretical and empirical benefits as those seen with WIS. These previous works in the linear domain concluded that weighting only the Bellman targets is less effective than applying importance weights to the entire loss function, which might even reduce variance (Graves & Ghiassian, 2022). Consequently, linear off-policy TD(0) naturally optimizes a superior objective without explicitly aiming to do so. Despite this evidence, deep learning approaches have predominantly continued to apply importance weights solely to the Bellman targets (Munos et al., 2016; Espeholt et al., 2018), as naively using the WIS loss with non-linear function approximations results in poor performance. Following the standard pattern for transferring ideas from tabular or linear function approximation settings to deep RL (Mnih et al., 2013; Lillicrap et al., 2015), a set of non-trivial design choices and extensions are required to achieve a scalable and performant WIS-based deep RL approach. We will discuss these design choices in Section 3.3 and their effectiveness in our experiments.

**Variance Analysis of Importance Sampling Estimators of the Bellman Error.**   When using the WIS loss, we can draw Monte Carlo samples from the joint state-action distribution. This mitigates the issue of having a single action per state, which can be limiting for a Equation 3 since requires separate estimates for both state and action distributions. Consequently, the WIS loss helps to reduce variance compared to the original and V-trace objectives. Moreover, one of the challenges with the use of importance sampling is that its estimator can exhibit high variance when the target and behavior policies significantly diverge. However, evaluating the variance of the importance sampling estimator is generally intractable (Munos et al., 2016; Koller & Friedman, 2009). Therefore, we consider a stateless MDP, i.e., a multi-armed bandit problem, as a simplified scenario. The critic's empirical losses for the base (Equation 2), WIS (Equation 3), and V-trace objective (Equation 4) can then be simplified for $N$ samples as

$$L_{\text{base}} = \left( V - \frac{1}{N} \sum_{a \sim \pi_b} \rho_a r(a) \right)^2 \qquad L_{\text{V-trace}} = \frac{1}{N} \sum_{a \sim \pi_b} \left( V - \left( (1 - \rho_a) V + \rho_a r(a) \right) \right)^2$$

$$L_{\text{WIS}} = \frac{1}{N} \sum_{a \sim \pi_b} \rho_a \left( V - r(a) \right)^2,$$

where we denote weights $\rho_a = \pi(a)/\pi_b(a)$. Taking the derivatives of the above losses with respect to the estimator $V$ and solving the derivative of zero for $V$ yields the following three estimators

$$\hat{V}_{\text{base}} = \frac{\sum_{a \sim \pi_b} \rho_a r(a)}{N} \qquad \hat{V}_{\text{V-trace}} = \frac{\sum_{a \sim \pi_b} \rho_a^2 r(a)}{\sum_{a \sim \pi_b} \rho_a^2} \qquad \hat{V}_{\text{WIS}} = \frac{\sum_{a \sim \pi_b} \rho_a r(a)}{\sum_{a \sim \pi_b} \rho_a}.$$

This simplified scenario illustrates the intuition that the WIS loss yields the self-normalized importance weighting estimator, while V-trace results in a squared self-normalized estimator and the base objective yields a standard importance weighting estimator. It is well established that the self-normalized estimator is more robust in the presence of extreme importance weights and often yields a lower variance than the standard importance weighting estimator in general machine learning (Koller & Friedman, 2009) and in RL settings (Swaminathan & Joachims, 2015; Futoma et al., 2020). However, the squared estimator from V-trace yields higher variances, as discussed in Appendix A.3.

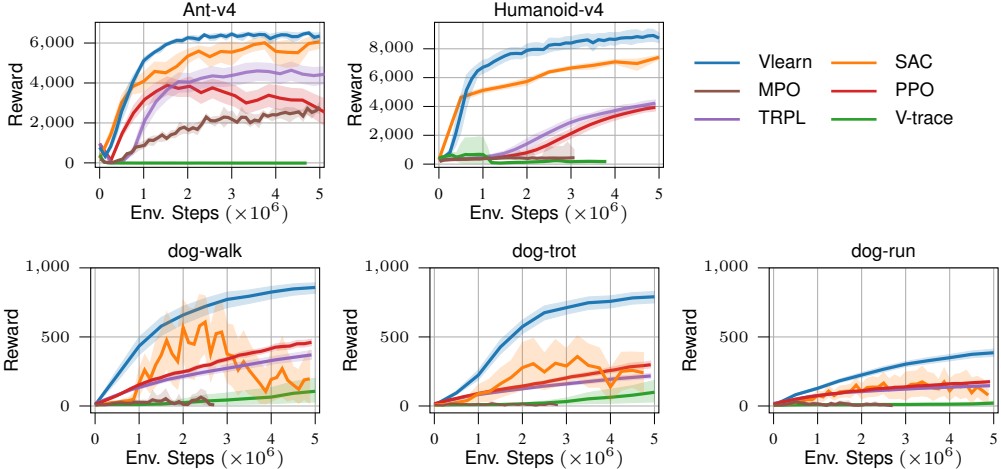

Figure 2: Mean over 10 seeds and 95% bootstrapped confidence intervals for the high-dimensional Gymnasium tasks, the 38-dimensional DMC dog tasks. Vlearn consistently achieves a better asymptotic performance for all tasks. For the dog tasks SAC and MPO even struggle to learn a consistent policy. Compared to V-trace our method is significantly more stable and achieves a better final performance.

Furthermore, unlike V-trace, each sample optimizes towards the Bellman target but has an impact on the total loss per step depending on its importance weight, as shown in Figure 1. The smaller importance weights mainly scale the gradient without causing a shift in optimizing to a different optimum, such as the current (target) value function. This approach makes learning the state-value function in an off-policy setting more stable and efficient.

### 3.3 DEEP OFF-POLICY POLICY LEARNING WITH STATE-VALUE FUNCTIONS

In order to leverage the previous findings about learning a state-value function, we need to ensure, we can also learn a policy. Conventional policy gradient techniques frequently use the gradient of the likelihood ratio and an importance sampling estimator to optimize the estimated Q-function. In particular, a more effective approach involves optimizing the advantage function, denoted as $A^\pi(s, a) = Q^\pi(s, a) - V^\pi(s)$. Adding the V-function as a baseline yields an unbiased gradient estimator with reduced variance. The optimization can then be formulated as follows

$$\max_\phi \hat{J}(\pi_\phi, \pi_b) = \max_\phi \mathbb{E}_{(s,a)\sim\mathcal{D}} \left[ \frac{\pi_\phi(a|s)}{\pi_b(a|s)} A^\pi(s, a) \right]. \tag{5}$$

In the on-policy case, the advantage values are commonly estimated via Monte Carlo approaches, such as general advantage estimation (Schulman et al., 2016). Yet, in our setting, we cannot reliably compute the Monte Carlo estimate and do not have a Q-function estimate (Degris et al., 2012). Further, the direct use of the V-function to improve the policy is not feasible. Nevertheless, using a replay buffer enables a strategy akin to the standard policy gradient. This approach enables us to utilize an off-policy estimate of the V-function, resulting in a significant improvement in sample efficiency. The advantage estimate is computed using the one-step return of our off-policy evaluated value function $A^\pi(s_t, a_t) = r_t + \gamma V_\theta^\pi(s_{t+1}) - V_\theta^\pi(s_t)$. Equation 5 can then be optimized by any policy gradient algorithm, e.g., *proximal policy optimization* (PPO) (Schulman et al., 2017) or *trust region projection layer* (TRPL) (Otto et al., 2021).

In this work, we use TRPL (Otto et al., 2021) as it has been shown to stabilize learning even for complex and high-dimensional action spaces (Otto et al., 2022; 2023). Unlike PPO, it provides a mathematically sound and scalable approach to enforce trust regions exactly per state. Moreover, TRPL allows us to use the constraint policy also during the value function update, which now requires importance sampling. PPO has only the clipped objective for the policy update, and using a clipped value function has been shown to potentially degrade performance (Engstrom et al., 2020).

TRPL efficiently enforces a trust region for each input state of the policy using differentiable convex optimization layers (Agrawal et al., 2019), providing more stability and control during training and at the same time reducing the dependency on implementation choices (Engstrom et al., 2020). Intuitively, the layer ensures the predicted Gaussian distribution from the policy network always satisfies the trust region constraint. This way, the objective from Equation 5 can directly be optimized as the trust region always holds. The layer receives the network's initial prediction for the mean $\boldsymbol{\mu}$ and covariance $\boldsymbol{\Sigma}$ of a Gaussian distribution and projects them into the trust region when either exceeds their respective bounds. This is done individually for each state provided to the network. The resulting Gaussian policy distribution, characterized by the projected parameters $\tilde{\boldsymbol{\mu}}$ and $\tilde{\boldsymbol{\Sigma}}$, is then used for subsequent computations, e.g., sampling and loss computation. Formally, as part of the forward pass, the layer solves the following two optimization problems for each state $\boldsymbol{s}$

$$\arg\min_{\tilde{\boldsymbol{\mu}}_s} d_{\text{mean}}\left(\tilde{\boldsymbol{\mu}}_s, \boldsymbol{\mu}(s)\right), \text{s.t.} \, d_{\text{mean}}\left(\tilde{\boldsymbol{\mu}}_s, \boldsymbol{\mu}_{\text{old}}(s)\right) \leq \epsilon_{\boldsymbol{\mu}}$$

$$\arg\min_{\tilde{\boldsymbol{\Sigma}}_s} d_{\text{cov}}\left(\tilde{\boldsymbol{\Sigma}}_s, \boldsymbol{\Sigma}(\boldsymbol{s})\right), \text{s.t.} \, d_{\text{cov}}\left(\tilde{\boldsymbol{\Sigma}}_s, \boldsymbol{\Sigma}_{\text{old}}(\boldsymbol{s})\right) \leq \epsilon_{\Sigma},$$

where $\tilde{\boldsymbol{\mu}}_s$ and $\tilde{\boldsymbol{\Sigma}}_s$ are the optimization variables for input state $\boldsymbol{s}$. The trust region bounds $\epsilon_{\mu}$, $\epsilon_{\Sigma}$ are for the mean and covariance of the Gaussian distribution, respectively. For the dissimilarities between means $d_{\text{mean}}$ and covariances $d_{\text{cov}}$, we use the decomposed KL-divergence. How to receive the gradients for the backward pass via implicit differentiation is described in Otto et al. (2021).

As the original V-trace algorithm is primarily used for off-policy corrections in distributed on-policy learning methods, it typically does not incorporate a target network. In contrast, it is common practice for off-policy methods (Haarnoja et al., 2018; Fujimoto et al., 2018; Abdolmaleki et al., 2018) to do so. This approach was also employed by V-trace's predecessor, Retrace (Munos et al., 2016), hence we adopt it for our use case as well. Additionally, we make use of further common improvements of actor-critic and policy gradient methods, including twin critics and delayed policy updates (Fujimoto et al., 2018), $\tanh$ squashing (Haarnoja et al., 2018), as well as advantage normalization (Schulman et al., 2017; Otto et al., 2021). While the overestimation bias is not a direct problem when using just state-value functions, we found twin critics beneficial in practice. We assume the twin network acts as a small ensemble that provides a form of regularization. Given there is no significant drawback and the widespread adoption of the twin network approach in other baseline methods, we chose to maintain it in our case as well. Finally, similar to prior works (Munos et al., 2016; Espeholt et al., 2018) we aim to reduce the variance by replacing the standard importance sampling ratio with truncated importance sampling $\min(\pi(a_t|s_t)/\pi_b(a_t|s_t), \epsilon_\rho)$.

Section 4.2 demonstrates the importance of employing a principled trust region approach and the necessity of these design choices for increased final performance. We call the final practical algorithm *Vlearn*, referring to its ability to learn from state-value functions. Pseudo-Code is given in Appendix B.

### 3.4 CONSIDERATIONS REGARDING OLD AND BEHAVIOR POLICY

For Vlearn, we keep track of three different policies: The current policy $\pi_\phi$, which is optimized, the old policy $\pi_{\text{old}}$, which is used as a reference for the trust region, and the behavioral policy $\pi_b$, which is needed for off-policy correction using importance sampling.

In particular, in an off-policy setting, the behavior policy $\pi_b$ is a mixture of all past policies that contributed to the replay buffer. However, storing and/or evaluating this would be expensive, so we assume that each sample belongs to only one mixture component, specifically the policy that originally created the action. Since we rely on importance sampling for the off-policy correction, we can simply store the (log-)probability for each action as part of the replay buffer to represent $\pi_b$ during training at minimal additional cost. This (log-)probability may then be used for importance sampling without the necessity of further computations.

Similar to most on-policy trust region methods, $\pi_{\text{old}}$ can simply be a copy of the main policy network from the previous iteration. Choosing $\pi_b$ for the trust region would be detrimental to performance because it would slow down the policy network update or, in the worst case, force the policy back to a much older and worse policy distribution. Conversely, we cannot use the copied $\pi_{\text{old}}$ as a behavior policy because the actual behavior policy $\pi_b$ can be arbitrarily far away, especially for older samples.

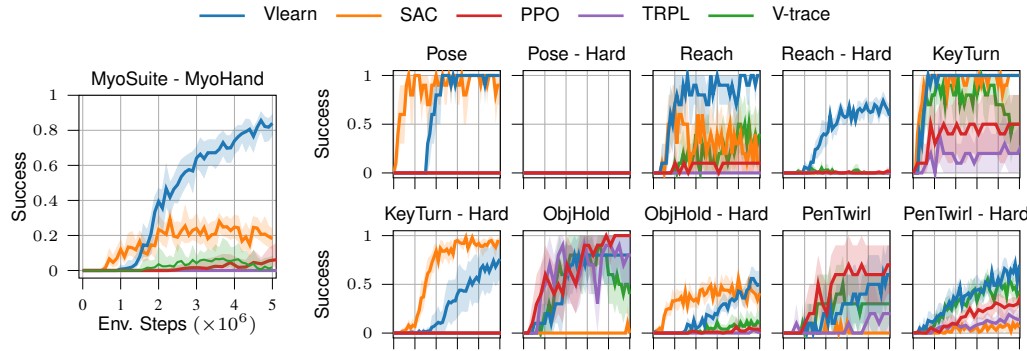

Figure 3: Performance on the 39-dimensional MyoHand from MyoSuite. **Left.** IQM and 95% bootstrapped confidence intervals for the aggregated performance over all 10 MyoHand tasks. **Right.** Mean over 10 seeds and 95% bootstrapped confidence intervals for the individual performances of all MyoHand tasks. While Vlearn does not outperform all baselines on all tasks, it performs well across all tasks and achieves the highest aggregated performance.

# 4 EXPERIMENTS

For our experiments, we evaluate Vlearn on various high-dimensional continuous control tasks from Gymnasium (Towers et al., 2023), DMC (Tunyasuvunakool et al., 2020) and MyoSuite (Vittorio et al., 2022). As baselines, we trained the standard off-policy methods SAC (Haarnoja et al., 2018) and MPO (Abdolmaleki et al., 2018) as well as PPO (Schulman et al., 2017) and the on-policy version of TRPL (Otto et al., 2021). In addition, we compare our method to V-trace (Espeholt et al., 2018) by replacing our lower bound objective from Equation 4 with the objective in Equation 3. All other components remain the same as in Vlearn. This comparison aims to highlight the difference in the placement of the importance ratios. To ensure a fair assessment, we want to eliminate any external factors that could influence the results and thus do not use n-step returns for both V-trace and Vlearn.

All methods are evaluated for 10 different seeds each, and their performance is aggregated using mean values and 95% bootstrapped confidence intervals for individual tasks as described in Agarwal et al. (2021). We maintain uniformity in the architecture of the policy and critic networks for all methods, incorporating layer normalization (Ba et al., 2016) before the first hidden layer. Hyperparameters are kept constant and only adjusted appropriately for the higher dimensional dog and MyoSuite tasks. Detailed hyperparameter information for all methods can be found in Appendix D.

## 4.1 HIGH-DIMENSIONAL CONTROL TASKS

As illustrated in Figure 2, for the two high-dimensional Gymnasium tasks, Ant-v4 and Humanoid-v4, Vlearn outperforms all other baselines. Across both environments, Vlearn demonstrates superior convergence speed and asymptotic performance. While the improvement is more subtle for Ant-v4 with its 8-dimensional action space, Vlearn achieves a 25% increase over SAC in the larger 17-dimensional action space of Humanoid-v4. Here, focusing solely on learning a state-value function proves to be a less complex and more effective approach than attempting to learn the full state-action value function. MPO is unable to achieve competitive performance for either Ant-v4 or Humanoid-v4.

The DMC (Tunyasuvunakool et al., 2020) dog tasks pose the most challenging locomotion problems in our evaluation, with a 38-dimensional action space modeling a realistic pharaoh dog. Consistent with the above Gymnasium tasks, Vlearn exhibits superior performance on these high-dimensional dog locomotion tasks. Although we found that SAC benefits from layer normalization for these tasks, it struggles to learn well-performing policies for all three movement types. While SAC improves for "easier" motions, its convergence remains highly unstable, and its final performance often falls below that of on-policy methods. Similar to the higher dimensional Gymnasium tasks, MPO is also unable to solve the dog tasks. In contrast, Vlearn learns reliably for all three distinct dog movement types.

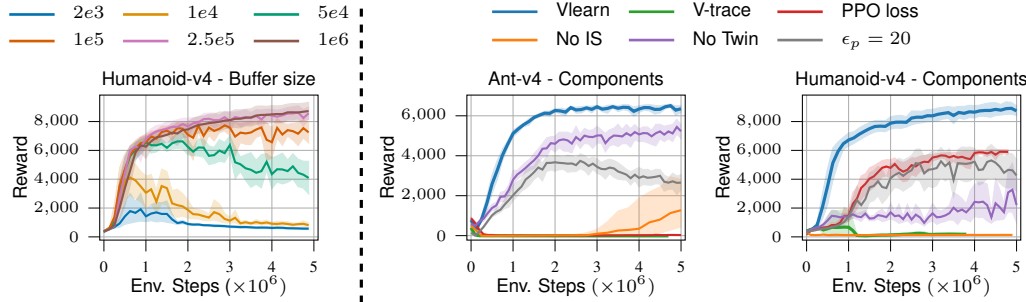

Figure 4: **Left.** Ablation study on the impact of replay buffer size on policy performance. For smaller replay buffers learning becomes unstable or does not converge, while larger sizes tend to lead to similar final performances. **Right.** Investigation in the importance of various design choices of Vlearn on Ant-v4 and Humanoid-v4.

From MyoSuite (Vittorio et al., 2022), we evaluated all tasks involving the 39-dimensional myoHand (see Figure 3). Vlearn shows consistently strong performance across a wide range of tasks, achieving the highest aggregated performance. Although it does not always outperform all baselines in specific cases, Vlearn tends to learn more stable behavior compared to SAC and V-trace. This stability is particularly evident in the easy versions of the tasks.

In direct comparisons between Vlearn and V-trace, Vlearn consistently demonstrates superior performance across all tasks. In particular, V-trace demonstrates minimal learning within a fixed number of environment interactions. For the dog tasks, the V-trace estimator fails to learn and is among the worst baselines, even falling behind on-policy methods. Our investigations revealed that while V-trace is eventually able to learn, it experiences a significant drop in performance after an extended training period. This discrepancy can be attributed to the sole difference between the two experiments, namely the V-trace objective. The key distinction between our objective and V-trace lies in how they handle situations where the importance ratio approaches zero and the position of the expectations. In our objective, samples get assigned a weight close to zero, minimizing their influence on the gradient. Conversely, V-trace attempts to bring these samples closer to the target network, potentially leading to performance degradation. Additionally, estimating the joint state-action expectation is typically more stable. This observation aligns with findings in Mahmood et al. (2014); Dann et al. (2014); Graves & Ghiassian (2022), which shows that importance sampling exclusively for the Bellman targets can lead to inferior performance. While they evaluated this only for the linear case, we found similar results for general nonlinear function approximation as shown in Figure 2.

## 4.2 IMPORTANCE OF DESIGN CHOICES

We investigate the effect of replay buffer size as well as the individual components of our method on the learning process. We trained multiple agents on the Gymnasium Humanoid-v4 with varying replay buffer sizes from $2e3$ up to $1e6$, for 10 seeds each. Note that while the original V-trace (Espeholt et al., 2018) relies on a relatively large replay buffer, the massively parallel computation used there implies that policy distribution differences are not as pronounced as in standard off-policy methods like Vlearn or SAC. As shown in Figure 4 (left), significant improvement occurs when transitioning from updating the policy and critical setting near an on-policy setting, using the smallest replay buffer size, to using a medium buffer. In particular, moving from a small buffer to a medium buffer yields significant benefits, while there is little difference between buffer sizes at the upper end of the range. In our experiments, we found that a replay buffer size of $5e5$ consistently produced optimal performance across all tasks and was used for the results in Figure 2, Figure 3, and Figure 5.

In order to analyze our design choices, we trained several variations of Vlearn for Ant-v4 and Humanoid-v4 to investigate the effects of each component (Figure 4 right). As a naive baseline, we removed importance sampling (*No IS*) and assumed that all samples in the replay buffer were from the current policy. Without the importance sampling correction, the agent was unable to learn in both environments. Replacing the TRPL policy loss with the clipped PPO loss (*PPO loss*), our results suggest that the heuristic trust region provided by PPO is insufficient for the off-policy case, where

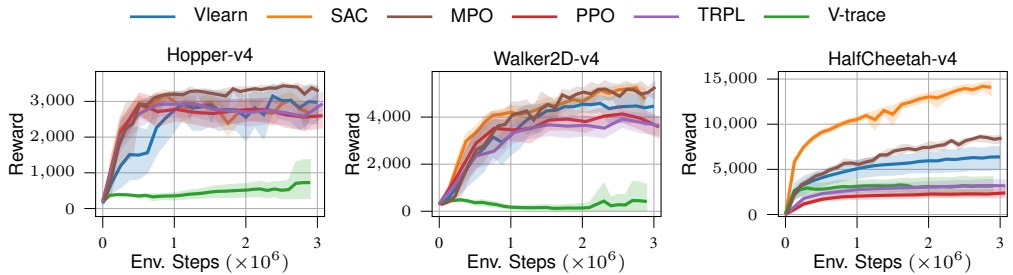

Figure 5: Mean over 10 seeds and 95% bootstrapped confidence intervals for the low-dimensional Gymnasium tasks. While SAC achieves the best performance on the HalfCheetah-v4, Vlearn still performs similar to MPO and achieves and equivalent performance for Hopper-v4 and Walker2D-v4. V-trace once again cannot make any meaningful progress in the off-policy setting.

stabilizing learning is even more important. While the PPO loss can achieve reasonable performance on Humanoid-v4, its asymptotic performance lags behind that of the agent using TRPL and is not able to learn for Ant-v4. For importance weight truncation, we followed previous work (Espeholt et al., 2018; Munos et al., 2016) by choosing $\epsilon_\rho = 1$ to reduce variance and potentially avoid exploding gradients. Similar to the PPO loss, we saw that learning becomes unstable for a larger truncation level ($\epsilon_\rho = 20$). Finally, we trained an agent without the twin critic networks (*No Twin*), which performs worse in both environments. We assume that the twin network can be seen as a small ensemble that provides a form of regularization that becomes more beneficial for higher dimensional problems.

### 4.3 Low-Dimensional Control Tasks

While the core benefit of our method is in high-dimensional action spaces, we ensure Vlearn is able to achieve competitive performance on lower-dimensional tasks. Generally, Vlearn performs well on the lower dimensional Gymnasium tasks shown in Figure 5. Its asymptotic performance is comparable to that of other baselines, except for SAC's performance on HalfCheetah-v4. While Vlearn outperforms on-policy methods on the HalfCheetah-v4, it exhibits a slower convergence rate compared to SAC and reaches a lower local optimum, similar to the on-policy methods. Generally, HalfCheetah-v4 seems to be an outlier that challenges the learning capabilities of all trust-region methods. Even MPO, employing a related KL regularization concept, does not match the performance of SAC. Moreover, the environment itself has shown extreme behavior in the past (Zhang et al., 2021). However, for the other lower dimensional tasks, Vlearn and MPO perform comparably to the remaining baselines.

### 5 Conclusion and Limitations

In this work, we have demonstrated an efficient approach for learning V-functions from off-policy data using an upper bound objective to subsequently update the policy network. Our approach not only ensures computational efficiency, but also demonstrates improved stability and performance compared to existing methods. Integrating this idea with TRPL yields an effective off-policy method that is particularly well-suited for scenarios involving high-dimensional action spaces.

Although our method excels in handling high-dimensional action spaces, it may still require a substantial amount of data to achieve optimal performance. Hence, sample efficiency remains a challenge that needs to be addressed further. In future work, we aim to further enhance the stability of off-policy state-value function-based methods and explore their potential benefits in tackling even higher-dimensional problems, such as those found in overactuated systems. Additionally, we interested in which ideas for Q-function-based methods also translate to the V-function only setting, such as ensembles (Chen et al., 2021), distributional (Bellemare et al., 2017) or categorical critics (Farebrother et al., 2024) as well as simplified ways to stabilize off-policy learning (Bhatt et al., 2024; Gallici et al., 2024). Furthermore, we are looking to extend our method to the realm of offline RL, which offers the opportunity to leverage pre-collected data efficiently, opening doors to real-world applications and minimizing the need for extensive data collection.

## 6 REPRODUCIBILITY STATEMENT

In our experiments, we utilized standard RL benchmarks and followed the evaluation protocols from Agarwal et al. (2021). The pseudo-code of our method, implementation details, and hyperparameters used in our experiments can be found in the Appendix. Readers can replicate our results by following the procedures outlined in these sections.

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

# A DERIVATION

We now show the proof of our new objective function. We start with a standard loss function for learning a V-value function for policy $\pi$ as defined in Equation 2. Then we derive its upper bound shown in Equation 4. For simplicity, we denote $\bar{V} = r(s, a) + \gamma V_{\bar{\theta}}(s')$.

## A.1 UPPER BOUND OBJECTIVE

$$\mathbb{E}_{s \sim d^\pi(s)} \left[ \left( V_\theta^\pi(s) - \mathbb{E}_{a \sim \pi(\cdot|s)} \bar{V} \right)^2 \right]$$

$$= \mathbb{E}_{s \sim d^\pi(s)} \left[ \left( V_\theta^\pi(s) - \mathbb{E}_{a \sim \pi_b(\cdot|s)} \left[ \frac{\pi(a|s)}{\pi_b(a|s)} \bar{V} \right] \right)^2 \right]$$

$$= \mathbb{E}_{s \sim d^\pi(s)} \left[ \left( \mathbb{E}_{a \sim \pi_b(\cdot|s)} \left[ \frac{\pi(a|s)}{\pi_b(a|s)} V_\theta^\pi(s) \right] - \mathbb{E}_{a \sim \pi_b(\cdot|s)} \left[ \frac{\pi(a|s)}{\pi_b(a|s)} \bar{V} \right] \right)^2 \right]$$

$$(V_\theta^\pi \text{ does not depend on a})$$

$$= \mathbb{E}_{s \sim d^\pi(s)} \left[ \left( \mathbb{E}_{a \sim \pi_b(\cdot|s)} \left[ \frac{\pi(a|s)}{\pi_b(a|s)} \left( V_\theta^\pi(s) - \bar{V} \right) \right] \right)^2 \right]$$

$$= \mathbb{E}_{s \sim d^\pi(s)} \left[ \left( \int \underbrace{\pi_b(a|s) \frac{\pi(a|s)}{\pi_b(a|s)}}_{\text{weight terms t}} \underbrace{\left( V_\theta^\pi(s) - \bar{V} \right)}_{x} da \right)^2 \right]$$

$$= \mathbb{E}_{s \sim d^\pi(s)} \left[ f \left( \int tx da \right) \right],$$

$$\leq \mathbb{E}_{s \sim d^\pi(s)} \left[ \int t f(x) da \right],$$

$$= \mathbb{E}_{s \sim d^\pi(s)} \left[ \mathbb{E}_{a \sim \pi_b(\cdot|s)} \left[ \frac{\pi(a|s)}{\pi_b(a|s)} \left( V_\theta^\pi(s) - \bar{V} \right)^2 \right] \right]$$

$$(\text{Jensen's inequality})$$

$$= \mathbb{E}_{s \sim d^\pi(s), a \sim \pi_b(\cdot|s)} \left[ \frac{\pi(a|s)}{\pi_b(a|s)} \left( V_\theta^\pi(s) - \bar{V} \right)^2 \right].$$

Here, we denote the convex function $f(x) = x^2$, the weight terms are in $[0, 1]$ and normalized, $\int t da = \int \pi_b(a|s) \frac{\pi(a|s)}{\pi_b(a|s)} da = \int \pi(a|s) da = 1$ and $d^\pi(s)$ is the stationary distribution induced by policy $\pi$. Therefore, one can estimate the loss using Monte Carlo samples from the joint state-action distribution

$$\sum_t \sum_j \frac{\pi(a_{t,j}|s_t)}{\pi_b(a_{t,j}|s_t)} \left( V_\theta^\pi(s_t) - \left( r_{t,j} + \gamma V_{\bar{\theta}}(s_{t+1,j}) \right) \right)^2.$$

## A.2 CONSISTENCY OF UPPER BOUND OBJECTIVE

We follow a similar derivation as in Neumann & Peters (2008) (for the case of state-action value function $Q(s, a)$) to prove the consistency between $L^*(\theta)$ and $L(\theta)$, i.e., the solution for minimizing

$L(\theta)$ is the same for the original objective $L^*(\theta)$.

$$L^*(\theta) = \mathbb{E}_{s \sim d^\pi(s)} \left[ \left( V_\theta^\pi(s) - \mathbb{E}_{a \sim \pi(\cdot|s)}[\bar{V}] \right)^2 \right]$$

$$= \mathbb{E}_{s \sim d^\pi(s)} [V_\theta^\pi(s)^2 - 2V_\theta^\pi(s)\mathbb{E}_{a \sim \pi(\cdot|s)}[\bar{V}] + \mathbb{E}_{a \sim \pi(\cdot|s)}[\bar{V}]^2]$$

$$L(\theta) = \mathbb{E}_{s \sim d^\pi(s)} \left[ \mathbb{E}_{a \sim \pi_b(\cdot|s)} \left[ \rho \left( V_\theta^\pi(s) - \bar{V} \right)^2 \right] \right]$$

$$= \mathbb{E}_{s \sim d^\pi(s)} [\mathbb{E}_{a \sim \pi_b(\cdot|s)}[\rho(V_\theta^\pi(s)^2 - 2V_\theta^\pi(s)\bar{V} + \bar{V}^2)]]$$

$$= \mathbb{E}_{s \sim d^\pi(s)} [V_\theta^\pi(s)^2 - 2V_\theta^\pi(s)\mathbb{E}_{a \sim \pi_b(\cdot|s)} \left[ \rho\bar{V} \right] + \mathbb{E}_{a \sim \pi_b(\cdot|s)} \left[ \rho\bar{V}^2 \right]]$$

$$(V_\theta^\pi \text{ does not depend on a})$$

$$= \mathbb{E}_{s \sim d^\pi(s)} [V_\theta^\pi(s)^2 - 2V_\theta^\pi(s)\mathbb{E}_{a \sim \pi(\cdot|s)} \left[ \bar{V} \right] + \mathbb{E}_{a \sim \pi(\cdot|s)} \left[ \bar{V}^2 \right]]$$

where $\rho = \pi(a|s)/\pi_b(a|s)$. The above results show that $L^*(\theta)$ and $L(\theta)$ are identical except for a constant term added, which remains independent of $V_\theta^\pi$.

### A.3 ANALYZING THE VARIANCE OF THE DIFFERENT IMPORTANCE SAMPLING ESTIMATORS

As shown in Section 3.2, for the bandit case the V-trace and weighted importance sampling estimators are given by

$$\hat{V}_{\text{V-trace}} = \frac{\sum_{a \sim \pi_b} \rho_a^2 r(a)}{\sum_{a \sim \pi_b} \rho_a^2} \quad \text{and} \quad \hat{V}_{\text{WIS}} = \frac{\sum_{a \sim \pi_b} \rho_a r(a)}{\sum_{a \sim \pi_b} \rho_a}$$

with the crucial difference that the importance weights are squared for V-trace. We can now consider the effective sample sizes for both, the standard self-normalized importance sampling estimator, $\rho_i x_i / \sum_j \rho_j$ and the squared version $\rho_i^2 x_i / \sum_j \rho_j^2$. They are given by

$$n_{\text{eff}}^\rho = \frac{\left( \sum_{i=1}^n \rho_i \right)^2}{\sum_{i=1}^n \rho_i^2} \quad \text{and} \quad n_{\text{eff}}^{\rho^2} = \frac{\left( \sum_{i=1}^n \rho_i^2 \right)^2}{\sum_{i=1}^n \rho_i^4},$$

respectively. Given that $\rho_i \geq 0$ with $\sum_{i=1}^n \rho_i$ being strictly positive and $n > 1$, it holds that $n_{\text{eff}}^\rho \geq n_{\text{eff}}^{\rho^2}$ (cf. Owen (2013) Chapter 9.3). Consequently, the variance in the squared case is always larger or equal to that of the standard estimator.

## B PSEUDO CODE

---

**Algorithm 1** Pseudo Code for Vlearn

---

1: Initialize policy $\phi$; critics $\theta_1, \theta_2$; target critics $\bar{\theta}_1, \bar{\theta}_2$
2: Initialize $\pi_{\text{old}} \leftarrow \pi_\phi$; trust region bounds $\epsilon_\mu, \epsilon_\Sigma$
3: Initialize replay buffer $\mathcal{D}$; truncation level $\epsilon_\rho$
4: **while** not converged **do**
5:     Collect sample $(s, a, r, s', \pi_\phi(a|s))$ and add to $\mathcal{D}$
6:     Sample batch $\mathcal{B} = \{(s, a, r, s', \pi_b(a|s))\}_{t=1\ldots K}$ from $\mathcal{D}$
7:     Get current policy $\pi_\phi(a|s)$ for all $s$ in $\mathcal{B}$
8:     Compute projected policy $\tilde{\pi}_\phi = \text{TRPL}(\pi_\phi, \pi_{\text{old}}, \epsilon_\mu, \epsilon_\Sigma)$ for all $s$ in $\mathcal{B}$
9:     Update critic network $i \in \{1, 2\}$ with gradient

$$\nabla_{\theta_i} \frac{1}{K} \sum_{\mathcal{B}} \min \left( \frac{\tilde{\pi}_\phi(a|s)}{\pi_b(a|s)}, \epsilon_\rho \right) \left( V_{\theta_i}^\pi(s) - \left( r + \gamma V_{\bar{\theta}_i}(s') \right) \right)^2$$

10:     **if** update policy **then**
11:         Compute advantage estimates $\hat{A} = r + \gamma \min_{i=1,2} V_{\theta_i}^\pi(s') - \min_{i=1,2} V_{\theta_i}^\pi(s)$
12:         Update policy with trust region loss $\nabla_\phi \left[ \frac{1}{K} \sum_{\mathcal{B}} \min \left( \frac{\tilde{\pi}_\phi(a|s)}{\pi_b(a|s)}, \epsilon_\rho \right) \hat{A} - \alpha \text{d}(\pi_\phi, \tilde{\pi}) \right]$
13:     **end if**
14:     $\bar{\theta}_i \leftarrow \tau\theta_i + (1-\tau)\bar{\theta}_i$ for $i \in \{1, 2\}$
15:     **if** update old policy for trust region **then**
16:         $\pi_{\text{old}} \leftarrow \pi_\phi$
17:     **end if**
18: **end while**

---

## C  ADDITIONAL RESULTS

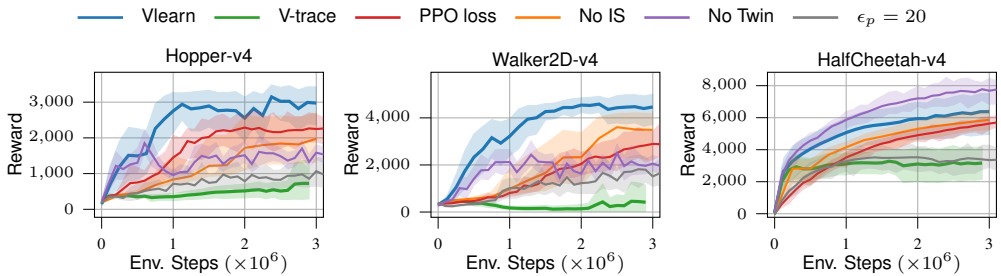

Figure 6: Performance of Vlearn without importance sampling (no IS) and when replacing TRPL with the PPO loss on low-dimensional Gymnasium tasks. Shown are the mean over 10 seeds and 95% bootstrapped confidence intervals. Similar to the high-dimensional setting, using the PPO loss or ignoring importance sampling altogether is not sufficient to ensure stable training and the performance suffers for most tasks.

## D  HYPERPARAMETERS

We conducted a random grid search to tune all hyperparameters for both Gymnasium[1] (Towers et al., 2023) and DMC[2] (Tunyasuvunakool et al., 2020), which are based on Mujoco[3] (Todorov et al., 2012). We included learning rates within the range $[10^{-5}, 5 \times 10^{-5}, 10^{-4}, 3 \times 10^{-4}, 5 \times 10^{-4}, 10^{-3}]$, mean and covariance bounds within $[10^{-5}, 10^{-4}, 5 \times 10^{-4}, 10^{-3}, 5 \times 10^{-3}, 10^{-2}, 5 \times 10^{-2}, 10^{-1}, 5 \times 10^{-1}]$, batch sizes of $[16, 32, 64, 128, 256, 512]$, policy update intervals of $[1, 2]$, and network sizes consisting of two hidden layers with $[64, 128, 256, 512]$ neurons each. All models were trained on an internal cluster on one Nvidia $V100$ for approximately 1-3 days, depending on the task. The best hyperparameter sets, as shown in Tables 1 and 2, were then trained for multiple seeds.

We found it is important of achieving a harmonious balance between the selected learning rate and the mean/covariance bound of the trust region. We observed that an aggressive learning rate does not pair well with a tight bound, and vice versa. Interestingly, when compared to other off-policy methods, we found that Vlearn and off-policy V-trace performed better with smaller batch sizes, and we did not observe any benefits from using warm start samples. In general, we noticed that the hyperparameters were a blend of configurations established from existing off-policy and on-policy methods. We attribute this to the structural similarities between our approach and both on-policy methods, such as PPO (Schulman et al., 2017) and TRPL (Otto et al., 2021), as well as off-policy methods, such as SAC (Haarnoja et al., 2018) and MPO[4] (Abdolmaleki et al., 2018).

---

[1]Published under MIT license at `https://github.com/Farama-Foundation/Gymnasium`

[2]Published under Apache-2.0 license at `https://github.com/google-deepmind/dm_control`

[3]Published under Apache-2.0 license at `https://github.com/google-deepmind/mujoco`

[4]Published under Apache-2.0 license at `https://github.com/google-deepmind/acme/`

Table 1: Hyperparameters for the Gymnasium (Towers et al., 2023) experiments in Figure 2 and Figure 5. The larger sample size for the the two on-policy methods is for the Humanoid-v4 experiments.

| | PPO | TRPL | Vlearn/V-trace | SAC | MPO |
|---|---|---|---|---|---|
| number samples | 2048/16384 | 2048/16384 | 1 | 1 | 1 |
| GAE $\lambda$ | 0.95 | 0.95 | n.a. | n.a. | n.a. |
| discount factor | 0.99 | 0.99 | 0.99 | 0.99 | 0.99 |
| $\epsilon_\mu$ | n.a. | 0.05 | 0.1 | n.a. | 1e-3 |
| $\epsilon_\Sigma$ | n.a. | 0.0005 | 0.0005 | n.a. | 2e-6 |
| optimizer | adam | adam | adam | adam | adam |
| updates per epoch | 10 | 20 | 1000 | 1000 | 1000 |
| learning rate | 3e-4 | 5e-5 | 5e-4 | 3e-4 | 3e-4 |
| epochs critic | 10 | 10 | n.a. | n.a. | n.a. |
| learning rate critic (and alpha) | 3e-4 | 3e-4 | 5e-4 | 3e-4 | 3e-4 |
| learning rate dual | n.a. | n.a. | n.a. | n.a. | 1e-2 |
| number minibatches | 32 | 64 | n.a. | n.a. | n.a. |
| batch size | n.a. | n.a. | 64 | 256 | 256 |
| buffer size | n.a. | n.a. | 5e5 | 1e6 | 1e6 |
| learning starts | 0 | 0 | 0 | 10000 | 10000 |
| policy update interval | n.a. | n.a. | 2 | 1 | 1 |
| polyak_weight | n.a. | n.a. | 5e-3 | 5e-3 | 5e-3 |
| trust region loss weight | n.a. | 10 | 10 | n.a. | n.a. |
| num action samples | n.a. | n.a. | n.a. | 1 | 20 |
| normalized observations | True | True | True | False | False |
| normalized rewards | True | False | False | False | False |
| observation clip | 10.0 | n.a. | n.a. | n.a. | n.a. |
| reward clip | 10.0 | n.a. | n.a. | n.a. | n.a. |
| critic clip | 0.2 | n.a. | n.a. | n.a. | n.a. |
| importance ratio clip | 0.2 | n.a. | n.a. | n.a. | n.a. |
| hidden layers | [64, 64] | [64, 64] | [256, 256] | [256,256] | [256,256] |
| hidden layers critic | [64, 64] | [64, 64] | [256, 256] | [256,256] | [256,256] |
| hidden activation | tanh | tanh | relu | relu | relu |
| initial std | 1.0 | 1.0 | 1.0 | 1.0 | 1.0 |

Table 2: Hyperparameters for the dog tasks from DMC (Tunyasuvunakool et al., 2020) and the MyoHand tasks from Myosuite (Vittorio et al., 2022) in Figure 2 and Figure 3, respectively.

|  | PPO | TRPL | Vlearn/V-trace | SAC | MPO |
|---|---|---|---|---|---|
| number samples | 16384 | 16384 | 1 | 1 | 1 |
| GAE $\lambda$ | 0.95 | 0.95 | n.a. | n.a. | n.a. |
| discount factor | 0.99 | 0.99 | 0.99 | 0.99 | 0.99 |
| | | | | | |
| $\epsilon_\mu$ | n.a. | 0.05 | 0.1 | n.a. | 1e-3 |
| $\epsilon_\Sigma$ | n.a. | 0.0005 | 0.0005 | n.a. | 1e-6 |
| | | | | | |
| optimizer | adam | adam | adam | adam | adam |
| updates per epoch | 10 | 20 | 1000 | 1000 | 1000 |
| learning rate | 3e-4 | 5e-5 | 1e-4 | 3e-4 | 3e-4 |
| epochs critic | 10 | 10 | n.a. | n.a. | n.a. |
| learning rate critic (and alpha) | 3e-4 | 3e-4 | 1e-4 | 3e-4 | 3e-4 |
| number minibatches | 32 | 64 | n.a. | n.a. | 1e-2 |
| batch size | n.a. | n.a. | 64 | 256 | 256 |
| buffer size | n.a. | n.a. | 5e5 | 1e6 | 1e6 |
| learning starts | 0 | 0 | 0 | 10000 | 10000 |
| policy update interval | n.a. | n.a. | 2 | 1 | 1 |
| polyak_weight | n.a. | n.a. | 5e-3 | 5e-3 | 5e-3 |
| trust region loss weight | n.a. | 10 | 10 | n.a. | n.a. |
| | | | | | |
| normalized observations | True | True | True | False | False |
| normalized rewards | True | False | False | False | False |
| observation clip | 10.0 | n.a. | n.a. | n.a. | n.a. |
| reward clip | 10.0 | n.a. | n.a. | n.a. | n.a. |
| critic clip | 0.2 | n.a. | n.a. | n.a. | n.a. |
| importance ratio clip | 0.2 | n.a. | n.a. | n.a. | n.a. |
| | | | | | |
| hidden layers | [64, 64] | [64, 64] | [512, 512] | [512,512] | [512,512] |
| hidden layers critic | [64, 64] | [64, 64] | [512, 512] | [512,512] | [512,512] |
| hidden activation | tanh | tanh | relu | relu | relu |
| initial std | 1.0 | 1.0 | 1.0 | 1.0 | 1.0 |

