# OpenReview forum: "Efficient Off-Policy Learning for High-Dimensional Action Spaces"
_ICLR.cc/2025/Conference — ICLR 2025 Poster_

### Official Review · Reviewer_urYk · 2024-10-25

**Soundness:** 3
**Presentation:** 3
**Contribution:** 2
**Rating:** 5
**Confidence:** 4

**Summary:**

The authors introduce Vlearn, an algorithm based on weighted importance sampling with state-value functions V, rather than state-action value functions Q. The authors argue this approach eliminates the complexity of learning action-values, particularly in domains with high dimensional action spaces and demonstrate performance benefits over SAC in MuJoCo environments.

**Strengths:**

- Vlearn provides a modern version of an old idea (weighted IS), including techniques from recent deep RL algorithms and showing competitive performance with deep RL algorithms.

- This is a (mostly) new flavor of algorithm in the context of deep RL and may open up the design space for further algorithms.

**Weaknesses:**

- Conceptual novelty is not high since the algorithm is based on well-known ideas in the literature. I think the “modern” version of these ideas has value, however it’s unclear how much value that is exactly.

- While the results are strong against SAC, SAC is definitely not state of the art in any of the tested domains. For example, TQC or TD7 for gym and BRO or iQRL for DMC are all model-free RL algorithms with much stronger performance than SAC. Consequently, it’s hard to get excited about these results since they underperform more modern algorithms, and I worry that the results may have overly benefited from environment specific tuning (dog) and may not generalize to more complex algorithms. For me to be convinced, I would like to see stronger results or covering more tasks/domains, to convince me that this algorithm is generally useful and not over-tuned.

**Questions:**

- I am surprised by the effectiveness of IS in a setting with high dimensional actions, where the ratios may have particularly high variance. Clipping can help avoid overly large values, but what about overly small values? Do the ratios remain close to one especially as the replay buffer grows and the behavior and current policies become more distinct?

- DMC also has a humanoid environment with a high dimensional action space. Does Vlearn also achieve a high performance in that task?

---

> ### Author Response · Authors · 2024-11-24
>
> Dear Reviewer urYk,
>
> We appreciate your thoughtful feedback and your evaluation of our work. We are glad that you recognize the value of providing a modern take on weighted importance sampling in the context of deep RL, and that our work might help expand the design space for RL algorithms.
> One of our goals was indeed to highlight how combining classical concepts with recent advancements in RL can yield effective algorithms, and we appreciate that this has been acknowledged.
>
> ## Novelty and Comparison to State-of-the-Art Algorithms
> While the foundational idea of weighted IS is indeed well-studied in tabular and linear settings, our primary contribution lies in its adaptation to the deep RL setting with V-functions and the demonstration of its competitiveness against established deep RL algorithms, which has not been explored in the literature.
> Furthermore, we provide theoretical insights regarding the variance behavior of different methods and the connection to linear and tabular methods.
>
> We do not seek to replace existing methods altogether, but rather, as the reviewer also mentions, to offer a new perspective and practical insights into deep off-policy RL that may inspire further innovation.
> Instead of relying on the usual Q-functions, it focuses exclusively on V-functions.
> A practical consequence of this is better suitability for high-dimensional action spaces, as this eliminates the need to learn over the joint state-action space, making the method less susceptible to the curse of dimensionality and more effective at updating the critic independent of the action.
> Our initial focus has been to demonstrate the general feasibility and benefits of such a setting on original baselines such as SAC, which typically serve as the basis for various improvements and variations.
> This involves providing theoretical insights into variance and methodology, while drawing parallels to the linear domain, and finally translating these ideas into a practical and functional algorithm.
> We therefore see methods such as TQC, TD7, BRO, iQRL, etc. as orthogonal work.
> In this sense, as also mentioned in our future work, we are interested which ideas from Q-function-based methods, such as distributional critics, critics ensembles, prioritized experience replay, network resets, quantization, or different critic/policy architecture choices, like BroNet, also translate to Q-function-free approaches.
> We tried to incorporate this motivation more clearly into our revised introduction.
>
> ## Generality and Overfitting Concerns
> In our revision, we added additional experiments on the $39$-dimensional MyoHand from MyoSuite using the same setting as for DMC's dog to cover a larger variety of tasks and provide evidence outside of the standard locomotion tasks.
> Similar to the initial submission, where we could show competitive performances on DMC's dog, Ant and Humanoid, our method is also able to learn consistently for MyoSuite and achieves a significantly better aggregated performance.
> Additionally, to mitigate potential over-tuning to specific environments, we followed a consistent training protocol across all environments and limited environment-specific hyperparameter tuning.
>
> > I am surprised by the effectiveness of IS in a setting with high dimensional actions, where the ratios may have particularly high variance. Clipping can help avoid overly large values, but what about overly small values? Do the ratios remain close to one especially as the replay buffer grows and the behavior and current policies become more distinct?
>
> We found that during learning the importance weights for older samples indeed decrease over time as the buffer grows.
> However, the impact of smaller importance weights is limited due to the structure of the loss that is induced by weighted importance sampling.
> As shown in Figure 1, the WIS loss behaves differently to e.g. the V-trace in that the loss is scaled, which means that the gradient for these samples is also scaled accordingly, limiting the potential impact they can have on learning stability.

---

> > ### Comment · Reviewer_urYk · 2024-11-24
> >
> > Thank you for the response.
> >
> > While I do appreciate the additional results, I think they suffer from the same problem I had originally, which is that SAC is not tuned for MyoSuite, so outperforming it is not significant. I'd be convinced by baselines intended for these benchmarks (e.g., TD-MPC2). For now, I will maintain my score.

---

> > > ### Author Response · Authors · 2024-11-28
> > >
> > > Thank you for your answer.
> > >
> > > We would like to emphasize that neither the goal nor the claim of this work is to achieve state-of-the-art performance.
> > > Instead, our goal is to provide a new perspective and practical insights into current off-policy deep RL, and potentially inspire further innovation based on the advantages presented in this work.
> > > We see our approach as a "vanilla" V-function method, and we wanted to illustrate its advantages over existing "vanilla" Q-function-based methods (SAC) and existing off-policy V-function methods (V-trace) - we have now added the results for the latter to MyoSuite as well, and Vlearn again clearly outperforms V-trace.
> > > In particular, we see that exploring the applications of Vlearn in a model-based setting, similar to TD-MPC2, is an interesting direction for future research.
> > >
> > > Regarding your concern about overtuning (for the dog task), we reused the exact same hyperparameters from the dog task across all MyoSuite experiments.
> > > Despite this, we were still able to achieve superior performance compared to SAC.

---

> > > > ### Comment · Reviewer_urYk · 2024-11-28
> > > >
> > > > While I agree it is not necessary to show state-of-the-art performance for this approach, SAC is a particularly weak baseline for these benchmarks since it was not tuned or intended for these benchmarks. This makes the results less significant.

---

### Official Review · Reviewer_mtMd · 2024-10-26

**Soundness:** 3
**Presentation:** 2
**Contribution:** 2
**Rating:** 6
**Confidence:** 4

**Summary:**

The paper proposes to use state-only value functions to model the critic during off-policy learning instead of using a state-action critic. It introduces a method to use importance sampling to correct for the distribution mismatch between data in the replay buffer and the policy being learned. The method has a theoretical justification and is applied to various high-dimensional domains.

**Strengths:**

- The technique is simple and relatively straightforward to adopt based on the presented pseudo-code.
- It is theoretically justified. Their proposed loss function upper-bounds the loss we actually want to minimize, and it shares the same solution as the loss we care about.
- Experiments are run over multiple seeds and results present confidence intervals.
- The presented ablations show the importance of different components of the paper’s algorithm.
- Figure 1 nicely illustrates some intuition about the method.

**Weaknesses:**

- The exact motivation of the paper is unclear to me. There are different thoughts scattered throughout, but I think these need to be consolidated to better understand what is actually being solved here. For example there are several seemingly related and unrelated comments that I think need to be consolidated: a) why are high-dimensional actions the actual issue? Our methods already handle high-dimensional states. If high-dimensional actions are an issue, it should be made clearer why. b) “Q-functions dependency on actions prevents it from updating actions that were not used while acting” (line 40); I agree this is an issue, but its only mentioned in passing. c) or is the main goal to scale off-policy algorithms?; if so there, are many other ideas and algorithms that need to be considered. d) or is the idea to just make an algorithmic improvement to V-trace? (line 173).
- Related to the above point. Since the motivation is unclear, I am not sure if the proposed baselines in the experiments are fair. The comparison to V-trace makes sense, but all the other algorithms are base/original implementations without any other techniques, so I would expect them to perform poorly anyway. For example, if the goal is to stabilize off-policy learning then there are plenty of algorithms that can be evaluated such as using layernorm [1] or batch norm [2], or removing the target network [1, 2] etc.
- The abstract makes a claim that the method improves upon exploration and exploitation. I dont think this is necessarily accurate. The experiments do not explicitly test for say exploration. It would be safer to say that it yields high-return agents.
- Lines 119 to 121 need a citation.
- The paper needs to fix its use of “target”. The RL literature typically uses it to refer to the target network and the target that the current estimate is trying to regress to. However, the paper uses target to refer to the current estimate as well (line 177), which makes parsing the paper challenging.
- Line 163 onwards discusses essentially the double sampling problem. A citation is needed here (see https://koppelresearch.wordpress.com/2018/09/13/double-sampling-a-quirk-of-reinforcement-learning/) and the term double sampling should be used. Also the way double sampling is presented here makes it seem that its an issue with equation (2) only and not equation (1), but its an issue for (1) as well.
- In my opinion, if the goal is for improved off-policy learning, the proposed algorithm introduces many more complicated pieces such as twin value function networks that are more complicated than methods that improve off-policy learning with simpler techniques (such as removing the target network and simply using layer norm [1]).

[1] SIMPLIFYING DEEP TEMPORAL DIFFERENCE LEARNING. Gallici et al. 2025.

[2] CrossQ: Batch Normalization in Deep Reinforcement Learning for Greater Sample Efficiency and Simplicity. Bhatt et al. 2024.

**Questions:**

I am having trouble wrapping my head around the fact that since each transition sample in the replay buffer has different associated log probabilities, learning will not be unstable. Typically, in importance sampling, the target and reference distributions are fixed. In this case, the target distribution is changing (since the policy is learning) and the reference is also changing since different transition samples in the buffer have different log probabilities. Do the authors have a sense of why this is not affecting learning? I would have expected instability. It would be interesting if there was an experiment to test this out.

---

> ### Author Response · Authors · 2024-11-24
>
> Dear Reviewer mtMd,
>
> Thank you for your detailed feedback and insightful comments.
> We are pleased that you found value in the simplicity, theoretical justification, and comprehensive experimental evaluation of our method.
> Your recognition of the ablation studies, confidence intervals, and the intuition provided by Figure 1 is particularly encouraging, as we aimed to present a clear and rigorous evaluation.
>
> ## Motivation
> We acknowledge that our paper could better consolidate its motivations.
> We do not seek to entirely replace existing methods, but rather to offer a new perspective and practical insights for deep off-policy RL.
> Rather than relying on the usual Q-functions, it focuses on V-functions exclusively.
> A practical consequence that follows is better suitability for high-dimensional action spaces since this eliminates the need to learn over the joint state-action space, making the method less susceptible to the curse of dimensionality, and more effective in updating the critic regardless of the action.
> Our initial focus was on demonstrating the general feasibility and benefits of such a setting against original baselines, like SAC, which typically serve as the basis for various improvements and variations.
> This includes providing theoretical insights regarding variance and methodology while drawing parallels to the linear domain, and lastly transferring these ideas into a practical and functional deep learning algorithm.
> As for V-trace, while our main goal is not necessarily to simply improve it, V-trace is the best directly comparable baseline for our setting.
> However, as discussed in the main paper, it naturally has certain shortcomings that can be avoided due to the different foundation of Vlearn.
> Finally, while we believe this shift away from state-action value learning to state value learning could potentially have implications for scaling RL, we do not consider this the main goal of this work.
> We tried to incorporate this motivation more clearly into our revised introduction.
>
> ## Baseline Comparison
> We see this work as a "vanilla" approach of an alternative direction to the main branch of current deep off-policy RL research.
> We think making changes to Vlearn, such as replacing target networks with BatchNorm/LayerNorm for stabilization, distributional critics, or other choices, are orthogonal to our line of work and are interested in our future work which ideas for Q-function-based methods also translate to Q-function-free approaches.
> For these reasons, we believe original baselines, such as SAC, provide a fair comparison.
> We have now also added additional evaluations on the $39$-dimensional MyoHand from MyoSuite using the same setting as for DMC's dog to cover a larger variety of tasks and provide evidence outside of the standard locomotion tasks.
> Similar to the initial submission, where we could show competitive performances on DMC's dog, Ant and Humanoid, our method is also able to learn consistently for MyoSuite and achieves a significantly better aggregated performance.
>
>
> ## Citations and Clarifications
> In the updated version, we revised the abstract and added citations and minor improvements as mentioned by the reviewer.
> We also did clarify, which "target" we are referring to.
> However, we wanted to point out that l.177 seems to be a misunderstanding, we do not refer to the current estimate as target.
> The intended meaning of this sentence was "[...] interpolating between the Bellman target and the current value function $(1 − \rho_t)V_\theta (s_t)$ or target value function $(1 − \rho_t)V_\bar\theta (s_t)$ in the 1-step case".
> This was an effort to generalize for cases with and without target network, but we agree this might be confusing and removed it.
>
>
> ## Importance Sampling and Replay Buffer
> Although our setting is off-policy, the importance sampling in general remains relatively similar to that of the on-policy setting.
> The changing target distribution is also present when updating the policy via importance sampling, e.g. in PPO.
> At the same time, the mini-batching, which is typically used in on-policy methods, is effectively acting as a small temporary replay buffer.
> As a result, we can expect similar instabilities as in those methods since the policy is updating rather slowly and is naturally bounded by leveraging a trust region via TRPL.
> However, one additional source of instability, which is significantly more relevant for the off-policy setting, is the "age"/staleness of samples, i.e. the discrepancy between the two policies used for importance sampling.
> As shown in our ablation results, leveraging the weighted importance sampling loss, importance weight clipping, etc. can alleviate these issues to ensure stable training.

---

> > ### Comment · Reviewer_mtMd · 2024-11-26
> >
> > Thanks to the authors for their response. I think generally things make sense and it is a nice insight. Contingent on improving the clarity of the motivation, I will increase the score. Thanks!

---

### Official Review · Reviewer_HaA8 · 2024-11-01

**Soundness:** 3
**Presentation:** 3
**Contribution:** 3
**Rating:** 8
**Confidence:** 4

**Summary:**

This paper introduces Vlearn, a novel off-policy policy gradient method for reinforcement learning. Extensive experimental results demonstrate that Vlearn achieves promising performance compared to existing approaches.

**Strengths:**

- Vlearn demonstrated promising performance compared to existing reinforcement learning algorithms in high-dimensional action spaces.
- Extensive experiments and ablation studies confirmed the effectiveness of Vlearn.
- Theoretical analysis provided insight into the motivation and design choices behind Vlearn.

**Weaknesses:**

Please refer to the questions part.

**Questions:**

I am curious about the distinction between “high-dimensional action space” and “low-dimensional action space.” In Section 4.3, experiments indicate that Vlearn performs less effectively than SAC. Could you elaborate on why Vlearn may not perform as well in low-dimensional action spaces?

---

> ### Author Response · Authors · 2024-11-24
>
> Dear Reviewer HaA8,
>
> We appreciate your positive feedback on our paper and are grateful for your insights.
> We are pleased to hear that you found our experimental results promising and that the theoretical analysis provided valuable context for our design choices.
>
> ## High-Dimensional vs Low-Dimensional Action Spaces
> High-dimensional action spaces often require more nuanced decision-making and pose a more complex learning problem in the case of Q-function-based approaches, such as SAC.
> They require learning over the joint state-action space, making them more susceptible to the curse of dimensionality.
> In contrast, low-dimensional action spaces typically only add a small amount of complexity when comparing V-function and Q-function learning, making it easier for algorithms like SAC to optimize effectively.
> Additionally, when only using a V-function in the off-policy setting importance sampling becomes necessary, which naturally adds additional variance to the training.
> From a simplified view, one could argue that this effectively boils down to a trade-off between the complexity of learning the critic vs the amount of additional variance.

---

### Official Review · Reviewer_ej3J · 2024-11-01

**Soundness:** 3
**Presentation:** 4
**Contribution:** 2
**Rating:** 8
**Confidence:** 3

**Summary:**

This work proposes a fully off-policy reinforcement learning algorithm based on policy gradient and importance sampling. Crucially, the algorithm does not attempt to estimate a state-action value function, but only a value function. As a consequence, the algorithm is well-suited for large action spaces. The authors build upon existing analysis in the linear case suggesting that the entire Bellman error should be reweighted, instead of the Bellman target alone. A simple analysis in bandit settings is provided, confirming that the chosen objective has less variance. The authors follow this idea in the deep reinforcement learning settings. The algorithm is evaluated in high-dimensional continuous control environments, showing that it is essential to combine the objective with a stable algorithm such as TRPL. The algorithm outperforms all baselines in large action spaces, and remains competitive in standard environments with smaller action spaces.

**Strengths:**

- The paper is clearly written, and the contribution is well outlined.
- Figure 1 is very helpful in describing the different behavior of Vlearn and Vtrace.
- The experimental evaluation is rigorous, involving multiple seeds and controlling for implementation differences and hyperparameter selection.
- Authors are transparent regarding limitations of the method on standard benchmarks (see Figure 4).

**Weaknesses:**

- This work opens an important question: why is a PPO-style loss insufficient for Vlearn, which thus requires TRPL? Figure 3 shows that changing the objective impacts performance in a drastic way. The answer provided ("the heuristic trust region provided by PPO is insufficient for the off-policy case", as well as lines 290-295) is not entirely satisfactory. A more detailed empirical or formal analysis would be very interesting.
- The contribution of this paper is largely in bringing ideas from the linear setting to deep RL. While this implies that the objective and the guarantees are not exactly novel, verifying that formal ideas can work in practice remains very valuable. In particular, the application of WIS in deep RL is, to the best of my knowledge, novel.

**Questions:**

- Can the authors comment on the first weakness?
- A lot of work seems to be necessary to ensure stable training of Vlearn. How is it expected to perform in noisy settings, or under non-stationary rewards? Would its performance be impact similarly to that of baselines, or more?
- In Appendix C, \citet commands should be replaced with \citep.

---

> ### Author Response · Authors · 2024-11-24
>
> Dear Reviewer ej3J,
>
> We appreciate the thoughtful feedback and the recognition of our work’s strengths. We are glad to hear that the clarity of our writing and the rigor of our experimental evaluation were well-received. Your comments on Figure 1 and the transparency regarding limitations are particularly encouraging.
>
> ## PPO-style Loss vs TRPL
> The trust region used by PPO effectively depends on the various implementation details present in PPO.
> This makes it more likely for the policy updates to be too large, i.e. violate the KL trust region, potentially leading to unstable learning (cf. the analysis in [1,2]).
> Paired with the off-policy setting and the increased variance due to the importance sampling, the overall gradient direction becomes less reliable, making these larger updates more risky.
> In Appendix C, we have also evaluated the PPO loss for the lower dimensional environments, for which the PPO loss works reasonably well.
> However, due to its heuristic nature, we found that it does not scale well to more complex problems.
>
> ## Performance in Noisy or Non-Stationary Settings
> While our current experiments focus on stationary settings, we anticipate that Vlearn may exhibit sensitivity to noise similar to other existing approaches, since this often affects all parts of training, including representation learning, credit assignment, etc.
> While trust regions such as those offered by TRPL can potentially alleviate some of these problems, they are unlikely to be sufficient.
>
> ### References
> [1] Fabian Otto, Philipp Becker, Ngo Anh Vien, Hanna Carolin Ziesche, and Gerhard Neumann. Differentiable trust region layers for deep reinforcement learning. In International Conference on Learning Representations, 2021.
> [2] Logan Engstrom, Andrew Ilyas, Shibani Santurkar, Dimitris Tsipras, Firdaus Janoos, Larry Rudolph,
> and Aleksander Madry. Implementation Matters in Deep Policy Gradients: A Case Study on PPO and TRPO. In International Conference on Learning Representations, 2020. URL http://arxiv.org/abs/2005.12729.

---

> ### Comment · Reviewer_ej3J · 2024-11-25
>
> Thank you for answering my questions. I will happily confirm my initial review.

---

### Author Response · Authors · 2024-11-24

Dear Reviewers,

We sincerely appreciate the detailed feedback and positive evaluations of our work.

We are pleased to hear that the experiments are considered thorough and that the theoretical analysis provides valuable insights into the motivation and design choices behind the algorithm.
We also appreciate that the reviewers value our transparency about the limitations of the method on standard benchmarks with low dimensional action spaces.
While most reviewers think that the proposed method is simple, relatively straightforward to adopt, and demonstrates promising performance while providing a "new flavor of algorithm in the context of deep RL and may open up the design space for further algorithms", some of the reviewers expressed concerns regarding the choice of baselines.

Our goal with Vlearn is to provide a new perspective and practical insights for deep off-policy RL by focusing exclusively on V-functions rather than the usual Q-functions.
Our initial focus was on demonstrating the general feasibility and benefits of such a setting for deep RL against foundational baselines, like SAC, which typically serve as the basis for various improvements and variations.
This includes providing theoretical insights regarding variance and methodology while drawing parallels to the linear domain and transferring these ideas into a practical and functional deep RL algorithm.
This approach is naturally better suited for high-dimensional action spaces as it avoids learning over the joint state-action space, making it less susceptible to the curse of dimensionality.
While weighted IS is a well-studied method in tabular and linear settings, our primary contribution lies in adapting it for deep RL with V-functions and demonstrating its competitiveness against established algorithms.

## Additional Experiments
We have added new experiments on the MyoSuite to improve our evaluations, demonstrate broader applicability, and address some of the reviewer's concerns.
These experiments cover $10$ tasks on the myoHand and demonstrate Vlearn's ability to handle high-dimensional environments beyond locomotion effectively.
In addition, Appendix C now includes the full ablation study for all lower-dimensional environments.

Thank you again for your constructive feedback and recognition of our work’s strengths.
We hope our revisions (highlighted in red) and new experiments effectively address your concerns.

---

> ### Author Response · Authors · 2024-11-28
>
> We want to thank all reviewers again for their time.
>
> We see our approach as a "vanilla" V-function method, and we wanted to illustrate its advantages over existing "vanilla" Q-function-based methods (SAC) and existing off-policy V-function methods (V-trace) - we have now added the results for the latter to MyoSuite as well, and Vlearn again clearly outperforms V-trace.

---

### Meta-Review · Area_Chair_JMzN · 2024-12-22

**Metareview:**

The paper proposes a new technique for offline RL that learns a state value function instead of state-action Q-function.  Combined with importance sampling, the new technique is a policy gradient technique that can leverage off-policy data.  The strength of this work is the simplicity of the proposed technique with a clear description, a theoretical analysis and good empirical evidence.  The weaknesses of this work include limited novelty since the proposed technique is an adaptation of prior work for linear settings to deep learning and comparison to baselines that are not the state of the art.  Nevertheless, the proposed technique will be of interest to the community and represents a valuable contribution.

**Additional Comments On Reviewer Discussion:**

The reviewers discussed the suitability of SAC as a baseline since it is not the state of the art for the test problems.  Ultimately, this does not take anything away from the proposed technique, it simply means that it is unclear whether the proposed technique advances the state of the art on the test problems or not.  The reviewers still felt that the proposed technique will be of interest to the community regardless.

---

### Decision · Program_Chairs · 2025-01-22

Accept (Poster)